# Genome Studies in Four Species of *Calendula* L. (Asteraceae) Using Satellite DNAs as Chromosome Markers

**DOI:** 10.3390/plants12234056

**Published:** 2023-12-02

**Authors:** Tatiana E. Samatadze, Olga Yu. Yurkevich, Firdaus M. Khazieva, Irina V. Basalaeva, Olga M. Savchenko, Svyatoslav A. Zoshchuk, Alexander I. Morozov, Alexandra V. Amosova, Olga V. Muravenko

**Affiliations:** 1Engelhardt Institute of Molecular Biology, Russian Academy of Sciences, 32 Vavilov St., 119991 Moscow, Russia; 2All-Russian Institute of Medicinal and Aromatic Plants, 7 Green St., 117216 Moscow, Russia

**Keywords:** *Calendula*, next-generation sequencing, repeatome, satDNAs, chromosome, fluorescence in situ hybridization

## Abstract

The taxonomically challenging genus *Calendula* L. (Asteraceae) includes lots of medicinal species characterized by their high morphological and karyological variability. For the first time, a repeatome analysis of a valuable medicinal plant *Calendula officinalis* L. was carried out using high-throughput genome DNA sequencing and RepeatExplorer/TAREAN pipelines. The FISH-based visualization of the 45S rDNA, 5S rDNA, and satellite DNAs of *C. officinalis* was performed on the chromosomes of *C. officinalis*, *C. stellata* Cav., *C. tripterocarpa* Rupr., and *C. arvensis* L. Three satellite DNAs were demonstrated to be new molecular chromosome markers to study the karyotype structure. Karyograms of the studied species were constructed, their ploidy status was specified, and their relationships were clarified. Our results showed that the *C. officinalis* karyotype differed from the karyotypes of the other three species, indicating its separate position in the *Calendula* phylogeny. However, the presence of common repeats revealed in the genomes of all the studied species could be related to their common origin. Our findings demonstrated that *C. stellata* contributed its genome to allotetraploid *C. tripterocarpa*, and *C. arvensis* is an allohexaploid hybrid between *C. stellata* and *C. tripterocarpa*. At the same time, further karyotype studies of various *Calendula* species are required to clarify the pathways of chromosomal reorganization that occurred during speciation.

## 1. Introduction

The genus *Calendula* L. (Asteraceae) includes 25 to 27 species of herbaceous plants and shrubs distributed mainly in the Mediterranean, Iran, Central Europe, Africa, and Asia [1]. Currently, various species of *Calendula* including *Calendula officinalis* L. (calendula or pot marigold), are widely used in the pharmaceutical, food, and cosmetic industries, and also as ornamental plants [2,3].

The flowers, leaves, and stems of various *Calendula* species contain flavonoids, xanthophylls, and carotenoids, essential oils, coumarins (scopoletin), and water-soluble polysaccharides (about 15%) [4]. The raw materials are rich in triterpenoid saponins (oleanolic acid glycosides), triterpene alcohols (ψ-taraxasterol, taraxasterol, faradiol, arnidiol, and heliantriol), and steroids [4,5]. The calendula flowers contain rutin which has antioxidant, anti-inflammatory, and anticarcinogenic activity [2,6]. The seeds are composed of about 20% fatty acids, and one of them, calendic acid, is the major (about 60%) constituent of calendula seed oil. Conjugated fatty acids are shown to be active substances for the treatment of obesity, and moreover, their antitumor effect has been revealed [7,8,9,10,11].

The genus *Calendula* is a taxonomically complex genus due to its high intraspecific morphological and karyological variability, and also its interspecific hybridization events [12,13,14,15,16,17,18,19,20,21,22]. In particular, the variations in its morphological characters, including seed heterocarpy, makes the identification of some species difficult [13,22]. Additionally, this genus is notable for its wide range of chromosome numbers across its species, with counts of 2n = 14 to ~85–88, which could be the result of multiple rounds of hybridization and genome duplication during speciation [14,18]. The basic chromosome numbers could be 7 (2n = 14, *C. stellata*), 8 (2n = 32, most perennial *Calendula* taxa from North Africa and Southwestern Europe), 9 (2n = 18, some perennial taxa from North Africa), 11 (2n = 44, *C. arvensis*), and 15 (2n = 30, *C. tripterocarpa*) [12,13,23,24,25]. It was also reported that *C. palaestina* and *C. pachysperma* had 2n = ±85 chromosomes, and those species were suggested to be autopolyploids of *C. arvensis* (2n = 44), whereas *C. arvensis* itself was assumed to be a tetraploid hybrid between two *Calendula* taxa with n = 15 and n = 7 [18]. At the same time, chromosome studies within the genus *Calendula* were carried out mainly using monochrome staining, which did not allow for the detailed analysis of the species’ karyotypes [12,13,17,23,26,27,28]. The chromosome numbers and plant morphology were mostly used to describe *Calendula* taxa and also determine their phylogenetic relationships [29,30]. In particular, *C. officinalis* was generally described as a tetraploid with a karyotype of 32 chromosomes (2n = 4x = 32) [31,32,33,34]. At the same time, the *C. officinalis* accessions with 2n = 28 chromosomes have also been detected [35,36,37]. FISH (fluorescent in situ hybridization) chromosome analysis has revealed four 35S/45S rDNA hybridization signals and two 5S rDNA signals in the *C. officinalis* karyotype with 2n = 28 chromosomes [36]. Later, in the 32-chromosome karyotype of *C. officinalis*, four major 45S rDNA, two 5S rDNA signals, and also two additional weak polymorphic 45S rDNA signals were detected [34,38], which supported the assumption of the allotetraploid origin of *C. officinalis*.

The investigation of genomic diversity within the genus *Calendula* using RAPD molecular markers revealed a high level of intraspecific genetic polymorphism among the studied genotypes [27], which could be related to the high level of spontaneous hybridization occurring between different taxa [13,22,39]. Such processes could have resulted in a wide range of intermediate variants of *Calendula* plants with different species names and belonging to different taxonomic groups [13]. Moreover, the phylogenetic analysis of *Calendula* based on nuclear and plastid DNA sequences (three non-coding chloroplast regions, atpIatpH, petLpsbE, and ndhFrpl32; internal transcribed spacers (ITS); and two putatively low-copy nuclear markers, Chs and A39) provided support for the division of this genus into annual and perennial polyploid complexes, and also the ‘multiple origins’ of most polyploid *Calendula* species and ‘a single origin’ of *C. officinalis* [40]. At the same time, the sequences of perennial *C. officinalis*, and also annual *C. stellata* Cav., *C. tripterocarpa* Rupr., and *C. arvensis* L., were found to belong to sister clades, which might be related to the presence of their common ancestors [40].

Another taxonomic revision of *Calendula* was made based on an evaluation of the nuclear DNA content, the genome size, number of chromosomes, and also the analysis of the life cycle of the plants. It was assumed that the differences in chromosome number as well as in the genome size and ploidy level could result from the high levels of hybridization, chromosome losses, and dysploidy [14,15]. At the same time, the evolutionary relationships and taxonomy within the genus *Calendula* are still unclear [14,40].

The FISH chromosome mapping of different molecular markers is one effective method to study karyotype structure. Using this approach, many questions in the systematics and phylogeny of plants, including *Calendula* species, have been successfully resolved. Recently, in various plant taxa, tandem DNA repeats have been used as effective chromosome markers to identify intra- and interspecific genome diversity, to detect chromosome rearrangements, and also to determine the species’ evolutionary pathways [41,42,43,44,45].

In the present study, the bioinformatic analysis of the *C. officinalis* genome DNA sequencing data was carried out to investigate its repeatome composition. The FISH-based mapping of 45S rDNA, 5S rDNA, and also the identified satellite DNA families (satDNAs) of *C. officinalis* was carried out on the chromosomes of *C. officinalis* and the related species *C. stellata*, *C. tripterocarpa*, and *C. arvensis* to determine their ploidy status and clarify the genome relationships of these species.

## 2. Results

### 2.1. Analyses of the Repetitive DNA Sequences Identified in the Genome of C. officinalis 

The repeatome analysis of *C. officinalis* showed that mobile genetic elements made up the majority of its repetitive DNA (Figure 1, Table 1). Retrotransposon elements (Class I), including Ty1-Copia and Ty3-Gypsy superfamilies, were highly abundant and represented 20.91% of the *C. officinalis* repeatome. In the Ty1-Copia superfamily (13.84%), SIRE (8.88%) and Angela (4.21%) were most abundant. Within the Ty3-Gypsy retroelements (5.8%), chromovirus Tekay (5.37%) dominated.

Moreover, the repeatome of *C. officinalis* contained large proportions of unclassified repeats (10.73%), and also ribosomal DNA (3.2%), and satellite DNA (3.92%). In the studied accession, four high-confidence and six low-confidence putative satDNAs were revealed using TAREAN (Figure 1, Table 1).

### 2.2. BLAST Analysis of the Identified SatDNAs 

According to BLAST, the five satDNAs (Cal 143, Cal 101, Cal 109, Cal 163, and Cal 187), identified in the genome of *C. officinalis* demonstrated sequence identity with satDNAs revealed in the genera *Glycine*, *Syngnathus*, *Patella*, *Pulicaria*, *Aphis, Cantharis*, *Harmonia,* and *Solea* (Table 2). A high sequence identity (84.9–94%) was revealed among the repeats Cal 2, Cal 5, and Cal 180, so in further FISH assays, we used only Cal 2, which showed the highest degree of sequence identity with both the Cal 5 and Cal 80 repeats.

### 2.3. Chromosomal Structural Variations

The cytogenetic analyses showed that the studied *Calendula* accessions have karyotypes with different numbers of chromosomes (*C. officinalis*, 2n = 32; *C. stellata*, 2n = 14; *C. tripterocarpa*, 2n = 30; and *C. arvensis*, 2n = 44) (Figure 2, Figure 3, Figure 4, Figure 5 and Figure 6).

In *C. officinalis*, the FISH-based mapping of the 45S rDNA and 5S rDNA revealed two satellite chromosome pairs bearing major clusters of 45S rDNA, and also one chromosome pair with small polymorphic clusters of 45S rDNA localized in the short arms. 5S rDNA clusters were observed in the short arms of one chromosome pair (Figure 2 and Figure 3). Different patterns of the chromosome distribution of the oligonucleotide Cal satDNA probes were revealed. Clusters of Cal 2, Cal 39, Cal 43, and Cal 163 were detected in the pericentromeric regions of most chromosomes. Additionally, clusters of Cal 43, and Cal 163 were detected in the intercalary and/or terminal regions of several chromosomes. Cal 101, Cal 103, Cal 109, and Cal 187 presented mixed clustered and dispersed localization (Figure 2 and Figure 3).

In the karyotypes of *C. stellata*, *C. tripterocarpa,* and *C. arvensis,* we examined the chromosome distribution of the satDNAs (Cal 2, Cal 39, Cal 43, and Cal 163) which demonstrated clustered localization on the chromosomes of *C. officinalis.* Among these four repeats, however, Cal 39 was localized dispersedly on the chromosomes of these *Calendula* species (Figure 4), and because of this, Cal 39 was not used for further analysis.

In the karyotype of *C. stellata*, large 45S rDNA clusters were detected in the short arms of two chromosome pairs. 5S rDNA signals were observed in the short arms of one chromosome pair (Figure 5). In the karyotype of *C. tripterocarpa*, 45S rDNA clusters of different sizes were revealed in the short arms of three chromosome pairs. 5S rDNA hybridization signals were observed in the short arms of four chromosome pairs (Figure 6). In the karyotype of *C. arvensis*, 45S rDNA clusters were found in the short arms of five chromosome pairs. 5S rDNA hybridization signals were observed in the short arms of another five chromosome pairs (Figure 6).

In *C. stellata*, large clusters of Cal 2 and Cal 43 were localized in the pericentromeric regions of all chromosomes. Additionally, Cal 2 and Cal 43 clusters were observed in the distal region of the chromosome pair 7 (short arms), and also small polymorphic clusters of Cal 43 were revealed in the terminal regions of the chromosome pairs 3, 4, and 5. Large Cal 163 clusters were revealed in the pericentromeric regions of all chromosomes and occupied the whole short arms of the chromosome pairs 1, 2, and 6 (Figure 5).

In *C. tripterocarpa*, clusters of Cal 2, Cal 43, and Cal 163 were localized in the pericentromeric regions of most chromosomes. Moreover, Cal 2 and Cal 43 clusters were detected at the end of the satellite on chromosome pairs 2, and in the distal and terminal regions of several chromosome pairs. Cal 163 clusters were revealed in the terminal regions of both arms of most of the chromosomes (Figure 6).

In *C. arvensis*, large Cal 2, Cal 43, and Cal 163 clusters were revealed in the pericentromeric regions of most chromosomes. Additionally, clusters of Cal 2, Cal 43, and Cal 163 were detected in the intercalary, distal, and/or the terminal regions of both arms in most of the chromosomes (Figure 6).

The analysis of the chromosome morphology and distribution patterns of the studied markers indicated that *C. officinalis* had an allotetraploid genome (2n = 4x = 32). The *C. tripterocarpa* genome also contained two subgenomes (2n = 4x= 30), one of which had significant similarity with the diploid genome of *C. stellata* (2n = 2x = 14). The second subgenome of *C. tripterocarpa* is derived from an unknown 16-chromosome ancestor, the genome of which differs from the genome of *C. stellata*. According to our results, *C. arvensis* had an allohexaploid genome, which resulted from a hybridization event between *C. stellata* × *C. tripterocarpa* (2n = 6x = 44). Thus, our findings allowed us to construct, for the first time, karyograms of the studied accessions of *C. officinalis*, *C. stellata*, *C. tripterocarpa*, and *C. arvensis* and clarify the ploidy status of these species (Figure 3, Figure 5 and Figure 6).

## 3. Discussion

Plant genomes contain large numbers of highly heterogeneous repetitive DNA including thousands or even tens of thousands of families which differ in motif length, copy number, and organization within the genome [46,47,48,49]. Transposable elements (TEs) are highly abundant and diverse fractions of plant genomes [50]. TEs can influence the genome organization and evolution since they can change their location and/or copy numbers [51,52]. Based on their structural characteristics and mode of replication, TEs are subdivided into two classes: class 1 (retrotransposons including LTR retrotransposons) and class 2 (DNA transposons) [51,52,53]. DNA transposons can move their locations in the genome with a ‘cut-and-paste’ mechanism [53]. LTR retrotransposons are the predominant group of TEs, constituting up to 75% of plant genomes [54,55,56,57]. LTR retroelements are the main contributors to the genome size variation within angiosperms since they can replicate and generate new copies with the ‘copy and paste’ mechanism, and also, they can be eliminated from the genome through both solo LTR formation and the accumulation of deletions [55,58,59,60]. These retroelements include the superfamilies Ty1-Copia and Ty3-Gypsy, which are further subdivided into a number of families, mostly specific to one or a group of closely related species [61]. In the present study, the repeatome analysis of *C. officinalis* also demonstrated that TEs made up the majority of its repetitive DNA. Retrotransposon elements, including the Ty1-Copia and Ty3-Gypsy superfamilies, were highly abundant. In the Ty1-Copia superfamily, the SIRE and Angela families were the most common, and in the Ty3-Gypsy, the Tekay chromovirus was predominant. This is typical for many species of vascular plants although the number of these retroelements can vary among taxa [44,45,62].

According to our results, ribosomal DNA made up about 3% of the *C. officinalis* genome. The sequences of the 45S (35S) rDNA and 5S rDNA, are known to be rather conserved in different eukaryotes, and hence, they are often used as FISH probes [36]. Previously, two major and also two minor clusters of 45S rDNA were revealed in the karyotype of *C. officinalis* [34,38]. In the studied accession of *C. officinalis*, we observed two major 45S rDNA clusters but only one minor cluster, which indicates the presence of intraspecific variability in this marker. In the karyotypes of *C. stellata*, *C. tripterocarpa*, and *C. arvensis*, the analysis of the chromosome morphology coupled with the chromosome distribution patterns of the 45S and 5S rDNA allowed us to reveal the similar chromosomes bearing these clusters.

The genome of the studied accession of *C. officinalis* contained substantial portions of (about 4 %) of satellite DNA. SatDNA sequences are considered to be the fast-evolving fractions of a plant repeatome, demonstrating divergence in both copy number and sequence even between closely related species [63]. SatDNAs can vary in a number of features including nucleotide composition, distribution, and abundance in plant genomes [64,65]. SatDNAs have a variable-length repeat unit (monomer) and usually form tandem arrays up to 100 Mb [66,67]. The sequences of the satellite monomers evolve concertedly via the process called ‘molecular drive’. Mutations are homogenized in a genome and become fixed in populations. The abundance of satDNA can vary within the plant genomes, and even between generations, resulting in high variability in the lengths of satellite arrays [68]. Some satDNA sequences, however, showed sequence conservatism for long evolutionary periods [69]. Although a high rate of genomic changes has been identified in different satellite DNAs, they can be either species-specific or common to a certain group of related species [68,69,70,71]. In the present study, according to BLAST, the sequence identity was revealed between two satDNAs identified in the *C. officinalis* genome and several DNA fractions of the *Pulicaria dysenterica* genome (Asteracea), indicating that these repeats are rather conservative within the Asteracea family. At the same time, the lack of data on the sequence homology between the identified satDNAs and the repeats of other *Calendula*-related taxa indicates the need for further studies of their repeatomes. The FISH-mapping of the satDNAs identified in the *C. officinalis* genome demonstrated the presence of common repeats in the chromosomes of all the studied *Calendula* species, which indicated the conservatism of these sequences within this genus. Considering the complexity of the taxonomy and phylogeny of this genus, the presence of common repeats in their genomes is important for clarifying the species relationships within the genus.

SatDNAs are often associated with heterochromatin and are localized in certain regions of chromosomes, which makes it possible to identify chromosome pairs in a karyotype, detect different chromosome rearrangements, estimate the range of chromosome variability and clarify species relationships [44,45,62,66,72]. The analysis of the chromosome distribution of the oligonucleotide satDNAs in the karyotypes of *C. officinalis*, *C. stellata*, *C. tripterocarpa,* and *C. arvensis* allowed us to detect three satDNAs (Cal 2, Cal 43, and Cal 163), which presented species-specific localization on the chromosomes of all the studied species and might be effective markers for the analysis of karyotypes within *Calendula*.

The taxa of the genus *Calendula* vary significantly in life cycle, morphology, genome size, and also chromosome number (2n = 14, 18, 30, 32, 44, and ~85–88) [18]. The morphological and karyological studies, coupled with the chromosome number variation, supported several hypotheses on the species origin and genomic diversity within the genus. In particular, a wide range of chromosome numbers within the genus might result from interspecific hybridization and polyploidization that occurred during speciation, which could lead to the appearance of a number of intermediate forms, complicating the taxonomy of this genus [12,13,14,15,18,21].

The intraspecific morphological variability of *C. officinalis*, as well as the presence of plants with different numbers of chromosomes (2n = 28, 32), were previously reported [31,32,33,34,35,36,37,38]. At the same time, the morphological similarities revealed between *C. officinalis* and some other *Calendula* taxa were also revealed, which could be related to interspecific hybridization events that might occur during speciation [19,21]. It was earlier assumed that *C. officinalis* with a karyotype of 32 chromosomes could be an allotetraploid (2n = 4x = 32) [31,32,33,34]. With the use of a set of different molecular chromosome markers, we demonstrated the allopolyploid status of the studied accession *C. officinalis* (2n = 4x= 32).

The genome size is considered to be an intrinsic property of a species, and intra- and interspecific variations in the genome size might reflect various evolutionary processes that occurred during speciation [64]. The genome sizes of *C. officinalis* (2.97 ± 0.08 pg/2C); *C. stellata* (2.11 ± 0.10 pg/2C), *C. tripterocarpa* (3.53 ± 0.12 pg/2C), and *C. arvensis* (5.20 ± 0.29 pg/2C) were estimated using flow cytometry [14,15]; and these genome sizes of the *Calendula* species were suggested to be related to their chromosome number, ploidy level, and also life cycle (perennial or annual) [15]. Moreover, genome size was used as one of the criteria for classifying species within the genus *Calendula*. Based on this criteria, three possible ploidy levels in *Calendula* were presented, namely, diploidy (e.g., *C. stellata*, 2n = 2x = 14), tetraploidy (e.g., in *C. officinalis*, 2n = 4x = 32 and *C. arvensis* 2n = 4x = 44), and octoploidy (e.g., *C. palaestina* and *C. pachysperma*, 2n = 8x = ±88). However, no inference about the ploidy level of *C. tripterocarpa* was made [15].

The karyotype of *C. stellata* is believed to be diploid (2n = 2x = 14) [18,73]. It was assumed that the karyotype of *C. stellata* could have resulted from aneuploidy or dysploidy, which led to the reduction in chromosomes from 2n = 18 to 2n = 14. [18,21,73]. In our study, we confirmed the chromosome number and ploidy status for the studied accessions of *C. stellata* (2n =2x= 14).

The origin of *C. tripterocarpa*, which is represented by three genotypes, 2n = 30 [17,18], 2n = 30 + 2B [73,74], and also 2n = 54 [75], is still controversial. Based on the chromosome number 2n = 30, *C. tripterocarpa* is considered to be either a diploid or a polyploid, and its ploidy level (2n = x? = 30) is still not fully understood [22]. It was previously assumed that *C. tripterocarpa* could be a hybrid between a hypothetical aneuploid or diploid ancestor of *C. stellata* with 2n = 16 chromosomes and *C. stellata* itself having 2n = 14 [13]. Our results mainly confirmed the earlier reported hypotheses of a hybrid origin for *C. tripterocarpa* [15,18,40]. Moreover, we demonstrated that *C. tripterocarpa* was an allotetraploid (2n = 4x = 30). At the same time, we noticed that only one of its subgenomes was similar to the *C. stellata* genome, and the second subgenome did not contain chromosomes from *C. stellata* genome. Our results indicated that an unknown 16-chromosome species was involved in the origin of the *C. tripterocarpa* genome. Therefore, further detailed cytogenomic studies of the related *Calendula* species having a karyotype with 2n = 16, 32 chromosomes are required.

The karyotype of the widespread and highly polymorphic annual species *C. arvensis* (2n = 44) was previously assumed to be a result of hybridization between *C. stellata* and *C. tripterocarpa*, followed by genome duplication [13,14,18]. In our study, we confirmed these chromosome numbers for the studied *C. arvensis* accession. We demonstrated that *C. stellata* contributed its genome to both *C. tripterocarpa* and *C. arvensis*. Therefore, *C. arvensis* is not a tetraploid, as previously suggested [15], but an allohexaploid (2n = 6x = 44), resulting from crossbreeding between diploid *C. stellata* and tetraploid *C. tripterocarpa* followed by chromosome duplication. 

Thus, further cytogenomic studies of various *Calendula* species are required to understand the functional and structural features of their genomes, as well as to clarify the pathways of chromosomal reorganization in their karyotypes during speciation.

## 4. Materials and Methods

### 4.1. Plant Material 

Seeds of *C. officinalis* cv. ‘Rajskij sad’ (K-304-212), *C. stellata* (K-479-10), *C. tripterocarpa* (K-366-21) and *C. arvensis* (K-304-211) were obtained from the collection of the Botanical Garden of the All-Russian Institute of medicinal and aromatic plants (Moscow, Russian Federation). For chromosome slide preparation, the seeds were germinated in Petri dishes on moist filter paper for 3–5 days at room temperature. Then, for genomic DNA extraction, the plants were grown in a greenhouse at 15 °C.

### 4.2. Genomic DNA Extraction and Sequencing

Genomic DNA of *C. officinalis* was isolated from young leaves with the use of the GeneJet Plant Genomic DNA Purification Kit (Thermo Fisher Scientific, Vilnius, Lithuania). The quality of the extracted DNA samples (DNAs) was checked with the Implen Nano Photometer N50 (Implen, Munich, Germany). The concentration and purification of the DNAs were assessed using the Qubit 4.0 fluorometer and Qubit 1X dsDNA HS Assay Kit (Thermo Fisher Scientific, Eugene, OR, USA).

Low-coverage whole genome sequencing was performed at the Beijing Genomics Institute (BGISeq platform) (Shenzhen, Guangdong, China) according to the NGS protocol for generating 5–10 million of paired-end reads of 150 bp in length, which was at least 1–2× of the coverage of the *C. officinalis* genome (1C = 726 Mbp) [14]. The raw sequencing data for *C. officinalis* (SAMN37557524) were uploaded to the NCBI (National Center for Biotechnology Information) BioProject database under accession number PRJNA1021511 (https://www.ncbi.nlm.nih.gov/bioproject/PRJNA1021511 (accessed on 27 September 2023)).

### 4.3. Sequence Analysis and Identification of DNA Repeats

The obtained genome sequences of *C. officinalis* were used for the genome-wide analyses, and also for the identification and characterization of major repeat families with the use of RepeatExplorer 2 and TAREAN pipelines [76,77]. The genomic reads were filtered by quality. One million high-quality reads were randomly selected for further analyses, which corresponds to 0.2× of a coverage of the *C. officinalis* genome (1C = 726 Mbp) [14]. This is within the limits of the genome coverage (0.01–0.50×) recommended by the developers of these programs [77]. RepeatExplorer/TAREAN was launched with the preset settings based on Galaxy platform (https://repeatexplorer-elixir.cerit-sc.cz/galaxy (accessed on 27 September 2023)). Initially, the preprocessing of the genomic reads was performed. Then, the reads were filtered in terms of quality using a cut-off of 10, trimmed, and filtered by size to obtain high-quality reads. Default threshold was explicitly set to 90% sequence similarity spanning at least 55% of the read length (in the case of reads differing in length, it applies to the longer one).

The sequence homology of the satDNAs identified in the genome of *C. officinalis* with repeats, which had been revealed earlier in other taxa, was estimated using BLAST (NCBI, Bethesda, MD, USA). Based on eight abundant satDNAs of *C. officinalis*, oligonucleotide FISH probes Cal 2, Cal 39, Cal 43, Cal 101, Cal 103, Cal 109, Cal 163, and Cal 187 (Table 3) were generated using the Primer3-Plus software [78].

### 4.4. Chromosome Slide Preparation

Root tips (0.5–1 cm long) were stored in ice water with 1 μg/mL of 9-aminoacridine (9-AMA) for 16–20 h. Then, the root tips were fixed in ethanol:glacial acetic acid (3:1) fixative for 3 days at 6–8 °C. The fixed roots were put into 1% acetocarmine solution (in 45% acetic acid) for 15–20 min. Then, a root tip was placed on the slide, the root meristem was cut from the tip cap, macerated in 45% acetic acid, covered with a cover slip, and a squashed chromosome preparation was made. After freezing in liquid nitrogen, the cover slip was removed. The slide was dehydrated in 96% ethanol and stored at −20 °C until use.

### 4.5. FISH Procedure

For the FISH assays, we used two wheat DNA probes: pTa71 containing 18S-5.8S-26S (45S) rDNA [79] and pTa794 containing 5S rDNA [80]. Both the probes were labelled directly with fluorochromes Aqua 431 dUTP (pTa71) and Red 580 dUTP (pTa794) (ENZO Life Sciences, Farmingdale, NY, USA) using nick translation according to manufacturers’ protocols. We also used eight oligonucleotide probes Cal 2, Cal 39, Cal 43, Cal 101, Cal 103, Cal 109, Cal 163, and Cal 187, which were produced and labelled directly with 6-FAM- or Cy3-dUTP in *Syntol* (Moscow, Russia).

Before the first FISH procedure, chromosome slides were pretreated with 1 mg/mL RNase A (Roche Diagnostics, Mannheim, Germany) in 2×SSC at 37 °C for 1 h. Then, the slides were washed three times for 10 min in 2×SSC, dehydrated through a graded ethanol series (70%, 85%, and 96%) for 3 min each and air-dried. Several sequential FISH procedures were performed with various combinations of these labelled DNA probes as described previously [62]. A total of 15 µL of hybridization mixture containing 40 ng of each labelled probe was added to each slide. Then, the slide was covered with a coverslip, sealed with rubber cement, denatured at 74 °C for 5 min, chilled on ice and placed in a moisture chamber at 37 °C for overnight. Then, the slide was washed in 0.1×SSC (10 min, 44 °C), twice in 2×SSC for 10 min at 44 °C, followed by a 5 min wash in 2×SSC and three 3 min washes in PBS at room temperature. Then, the slide was dehydrated through a graded ethanol series for 2 min each and stained with DAPI (40,6-diamidino-2-phenylindole) dissolved (0.1 µg/mL) in Vectashield mounting medium (Vector Laboratories, Burlingame, CA, USA). After documenting FISH results, the chromosome slide was washed in distilled water for 10 min, and the sequential FISH procedure was conducted on the same slide.

### 4.6. Chromosome Analysis

The chromosome slides were analyzed using the epifluorescence microscope (Olympus BX61) with the standard narrow band pass filter set and UPlanSApo 100/1.40 oil UIS2 objective (Olympus, Tokyo, Japan). Chromosome images were captured with a monochrome CCD (charge-coupled device) camera (Snap, Roper Scientific, Tucson, AZ, USA) in grayscale channels. Then, the images were pseudo-colored, and processed with Adobe Photoshop 10.0 (Adobe Systems, Birmingham, AL, USA) and VideoTesT-FISH 2.1 (IstaVideoTesT, St. Petersburg, Russia) software. For each species sample, at least five plants and 15 metaphase plates were studied. Chromosome pairs in karyotypes were identified according to the chromosome size and morphology, localization of chromosome markers, and also the cytological nomenclature proposed previously [34].

## 5. Conclusions

New effective chromosomal markers (Cal 2, Cal 43, and Cal 163) were detected for the analysis of *Calendula* karyotypes. Our results show that the karyotype of *C. officinalis* differs from karyotypes of *C. stellata*, *C. tripterocarpa* and *C. arvensis*, however, the presence of common repeats in their genomes could be related to their common origin. The ploidy status of *C. officinalis* was specified as tetraploid. Our findings demonstrate that diploid *C. stellata* has contributed its genome to the allotetraploid *C. tripterocarpa*, and *C. arvensis* is an allohexaploid hybrid between *C. stellata* and *C. tripterocarpa*. Our approach could be useful for further cytogenomic studies of various *Calendula* species.

## Figures and Tables

**Figure 1 plants-12-04056-f001:**
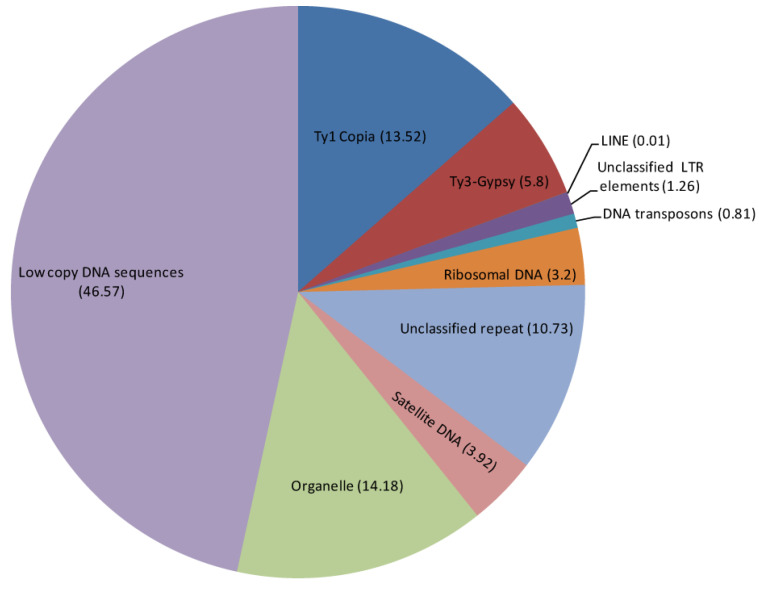
Types of highly and moderately repeated DNA sequences in the *Calendula officinalis* genome. A TE proportion of each repeat type or family is shown inside parenthesis.

**Figure 2 plants-12-04056-f002:**
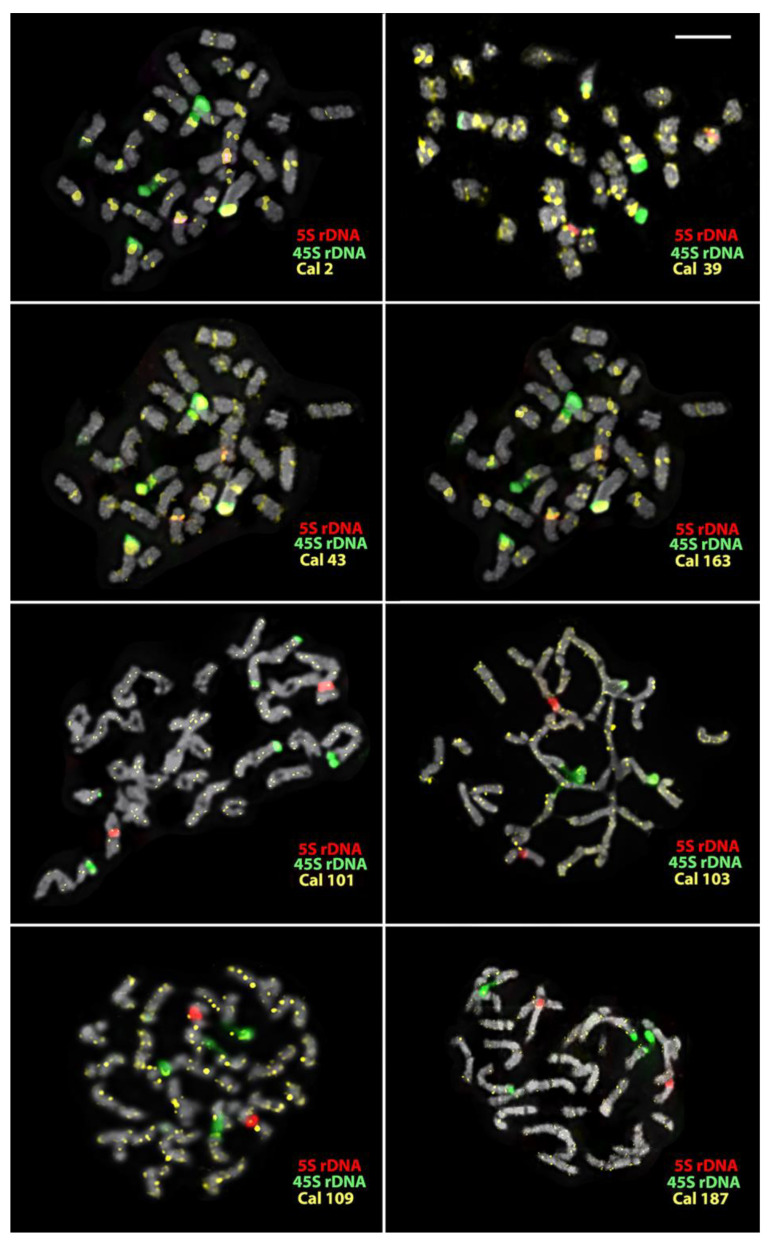
Localization of the studied molecular cytogenetic markers on chromosomes of *Calendula officinalis*. Merged images after multicolor FISH with 45S rDNA, 5S rDNA, Cal 2, Cal 39, Cal 43, and Cal 163. The names of the probes and their pseudocolors are indicated on the lower right of each metaphase plate. Scale bar—5 µm.

**Figure 3 plants-12-04056-f003:**
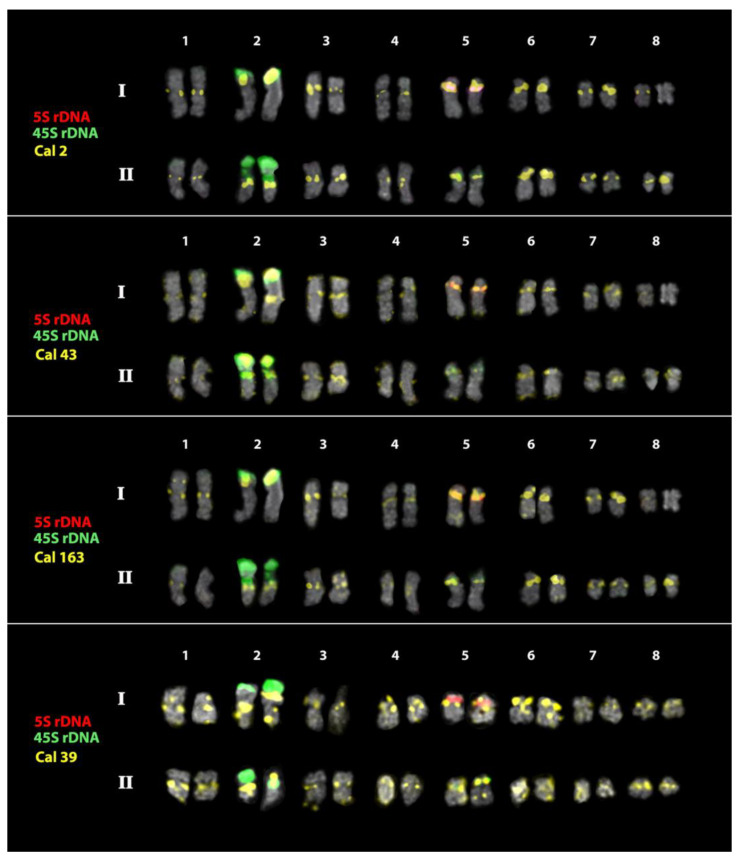
Karyograms of *C. officinalis* after multicolor FISH with 45S rDNA, 5S rDNA, Cal 2, Cal 43, Cal 163, and Cal 39. The same metaphase plates are shown as in Figure 2. I–II—subgenomes. The names of the probes and their pseudocolors are indicated on the left.

**Figure 4 plants-12-04056-f004:**
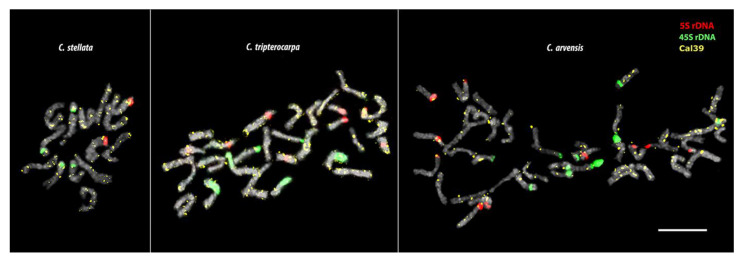
Metaphase plates of *C. stellata, C. tripterocarpa* and *C. arvensis* after multicolour FISH with 45S rDNA (green), 5S rDNA (red), and Cal 39 (yellow). Scale bar—5 µm.

**Figure 5 plants-12-04056-f005:**
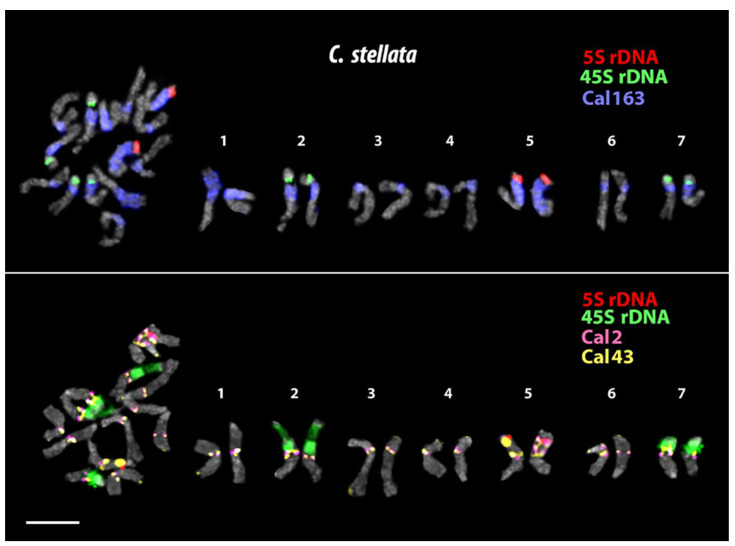
Localization of 45S rDNA(green), 5S rDNA (red), Cal 2 (pink), Cal 43 (yellow), and Cal 163 (blue) on chromosomes of *C. stellata*. Scale bar—5 µm.

**Figure 6 plants-12-04056-f006:**
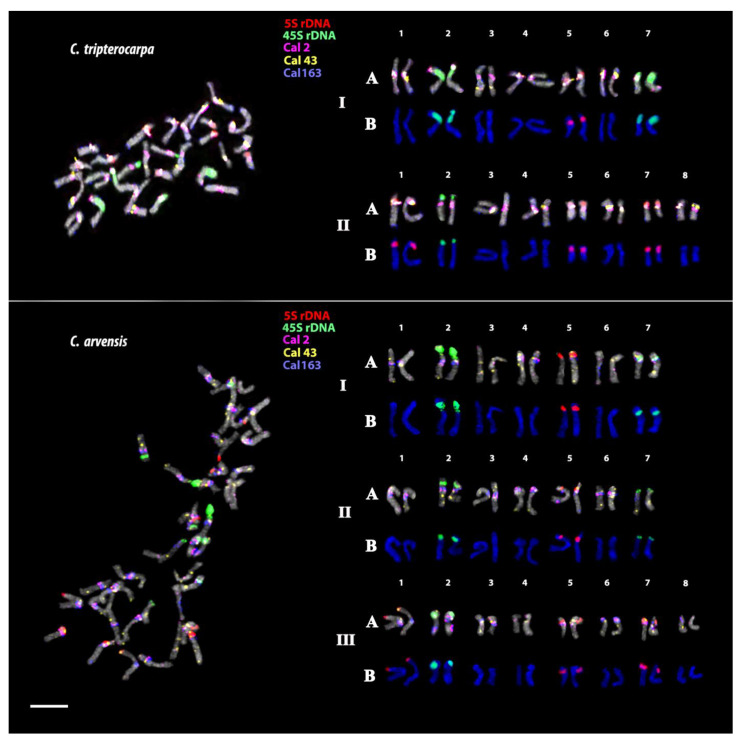
Localization of (A) 45S rDNA (green), 5S rDNA (red), Cal 2 (pink), Cal 43 (yellow), Cal 163 (blue), and DAPI-staining (grey) and also (B) 45S rDNA (green), 5S rDNA (red), and DAPI-staining (blue) on chromosomes of *C. tripterocarpa* and *C. arvensis*. I–III—subgenomes. Scale bar—5 µm.

**Table 1 plants-12-04056-t001:** Proportions of major repetitive DNA repeats identified in the genome of *Calendula officinalis*.

Repeat Name	Genome Proportion (%)
Retrotransposons (Class I)	20.91
Ty1 Copia	13.84
Angela	4.21
Bianca	0.19
SIRE	8.88
TAR	0.24
Unclassified Ty1 copia elements	0.32
Ty3-Gypsy	5.80
Non-chromovirus Athila	0.24
Non-chromovirus Tat-Retand	0.19
Chromovirus Tekay	5.37
LINE	0.01
Unclassified LTR elements	1.26
Transposons (Class II)	0.81
Cacta	0.07
hAT	0.02
MuDR_Mutator	0.03
PIF_Harbinger	0.39
Helitron	0.30
rDNA	3.20
Unclassified repeats	10.73
DNA satellite	3.92
Organelle	14.18
Putative satellites	4 high confident
	6 low confident

**Table 2 plants-12-04056-t002:** Comparison of the satDNAs identified in the genome of *Calendula officinalis* with the available data.

Tandem Repeat/ Genome Proportion, %	Repeat Length, bp	BLAST Homology with Other Identified Cal Repeats	BLAST Homology (Available NCBI Data)
Cal 2/1.2	90	88.9% identity with Cal 5, 94% with Cal 80	not found
Cal 5/1.1	136	88.9% identity with Cal 2, 84.9% with Cal 80	not found
Ca 43/0.51	267	not found	*Glycine max* cultivar Williams 82 chromosome 19/88.1%; CP126444.1 *Glycine max* cultivar Williams 82 chromosome 17/86.4%; CP126442.1
Cal 80/0.25	90	94% identity with Cal 2, 84.9% with Cal 5	not found
Cal 39/0.54	14	not found	not found
Cal 101/0.1	90	not found	*Syngnathus acus* genome assembly chromosome 9/91.7%; OX411224.1
Cal 103/0.093	32	not found	not found
Cal 109/0.074	1893	not found	*Pulicaria dysenterica* genome assembly chromosome 6/80%; OX359293.1 *Pulicaria dysenterica* genome assembly chromosome 7/85.5%; OX359294.1 *Patella depressa* genome assembly chromosome 3/85.7%; OX419717.1
Cal 163/0.019	83	not found	*Aphis gossypii* genome assembly chromosome 3/96.3%; OU899036.1 *Cantharis lateralis* genome assembly chromosome 5/92.5%; OY720628.1 *Harmonia axyridis* genome assembly chromosome 3/88.4%; OU611929.1
Cal 187/0.013	375	not found	*Pulicaria dysenterica* genome assembly chromosome 5/70.4%; OX359292.1 *Pulicaria dysenterica* genome assembly chromosome 9/69.2%; OX359296.1 *Solea solea* genome assembly chromosome 8/92.3%; OY282541.1

**Table 3 plants-12-04056-t003:** List of the generated oligonucleotide FISH probes.

Oligonucleotide Probes	Sequences of the Oligonucleotide Probes
Cal 2	ATAAGTATCCATTTTAAACCGTAATAGGTGTCCATAACCCATACGAATGGCCC
Cal 39	GCTCAAGGCTCAAG
Ca 43	AAAGGCCATAACTTTTGGCTCGGGTCTCCGTTT
Cal 101	AAATCACGAAGCACATGTGCTTCATAAAGCCAAGCACAT
Cal 103	ACATATAGTCAAGATTAATCATCGATGATTAA
Cal 109	CTAACAATCTCCCCCTATCTGATGATAACAAATGAATATGTTTAA
Cal 163	CGTTAAATTCAATATTTTTCTGGAATTTTCCAAGATTCCTTGAATTTATAACCTTAAATATGTATT
Cal 187	CAAGTGGGAGGGAAAACGTATAAGAGCACCAAGGTGTTTGGAAATGAA

## Data Availability

All data generated or analyzed during this study are contained within the article.

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
