# Peer review of "Genome Studies in Four Species of Calendula L. (Asteraceae) Using Satellite DNAs as Chromosome Markers"

_plants, 2023, doi:10.3390/plants12234056_

Round 1

Reviewer 1 Report

Comments and Suggestions for Authors

I like the FISH conducted in this study; the karyotype analysis is well done. The manuscript is well-written.

I only have a few suggestions for the authors:

The authors uncovered that the C. officinalis karyotype differed from the other three species' karyotypes and suggested its separate position in Calendula phylogeny. If they can, shall the authors cite some references that have conducted phylogenetic analyses in Calendula to support their hypothesis?

Author Response

Response to the Reviewer's Comments and Suggestions

Reviewer 1

I like the FISH conducted in this study; the karyotype analysis is well done. The manuscript is well-written.

I only have a few suggestions for the authors:

The authors uncovered that the C. officinalis karyotype differed from the other three species' karyotypes and suggested its separate position in Calendula phylogeny. If they can, shall the authors cite some references that have conducted phylogenetic analyses in Calendula to support their hypothesis?

Answer: We would like to thank the reviewer for helpful comments on our manuscript.

Answer: This has been corrected. The reference [40] has been cited. Lines 81-82.

Reviewer 2 Report

Comments and Suggestions for Authors

In this manuscript the authors propose a repeatome analysis of a valuable medicinal plant Calendula officinalis L. using high throughput DNA sequencing and RepeatExplorer/TAREAN pipelines. New effective chromosomal markers important for studies on karyotype diversity within the genus Calendula were detected, shedding new light on the phylogeny of Calendula and the different ploidy levels studied.

The article in its simplicity is well constructed, the results are aimed at demonstrating what has been achieved. However, the authors should review the article in its entirety and elaborate more on the written part: sometimes it abounds in short sentences, sometimes used to introduce the next sentence, but which ultimately represents a repetition. Even the care of what is reported is an indication of an elaboration of the research.

Best Regards.

Author Response

Response to the Reviewer's Comments and Suggestions

Reviewer 2

In this manuscript the authors propose a repeatome analysis of a valuable medicinal plant Calendula officinalis L. using high throughput DNA sequencing and RepeatExplorer/TAREAN pipelines. New effective chromosomal markers important for studies on karyotype diversity within the genus Calendula were detected, shedding new light on the phylogeny of Calendula and the different ploidy levels studied.

The article in its simplicity is well constructed, the results are aimed at demonstrating what has been achieved. However, the authors should review the article in its entirety and elaborate more on the written part: sometimes it abounds in short sentences, sometimes used to introduce the next sentence, but which ultimately represents a repetition. Even the care of what is reported is an indication of an elaboration of the research.

Answer: We are very grateful to the reviewer’s comments and thoughtful suggestions.

Answer: We have revised the manuscript according to the Reviewer’s comments (shown in green).

Reviewer 3 Report

Comments and Suggestions for Authors

Reviewer : The manuscript "Genome Studies in Four Species of Calendula L. (Asteraceae)

Using Satellite DNAs as Chromosome Markers" by Samatadze, et al., who carried out the bioinformatic analysis of the C. officinalis DNA sequencing data, also clarified ploidy status and genome relationships of four Calendula species based on FISH mapping of 45S rDNA, 5S rDNA, and the identified satellite DNA families. This study provides a basis for the classification of Calendula L. (Asteraceae) . And the manuscript is well written and thoroughly described and it can find interest among the researchers in this field. However, there are some comments or questions before its possible publication as follows:

1. The language used in this manuscript needs attentions, it should be edited by a native English-speaking expert.

2. There are too many keywords, which need to be streamlined and avoided being the same as the title words. At the same time, I think it is necessary to use complete words when abbreviations appear for the first time in the text, like NGS, FISH.

3. Figure 1 is not a good form of expression, it is unsightly and incomprehensible, so it is suggested to replace it with other drawing forms.

4. In Table 1, all data need to be kept two decimal places.

5. In Table 2, the format of the three-line table is incomplete, please modify and supplement it.

6. The title of 2.2 is not specific enough to know the content of this part directly through the title.

7. Although the manuscript cited more than 80 reference, more than half of them are outdated, which need to be updated and the preface and discussion part should be carefully revised.

Comments on the Quality of English Language

The language used in this manuscript needs attentions, it should be edited by a native English-speaking expert.

Author Response

Response to the Reviewer's Comments and Suggestions

Reviewer 3:

The manuscript "Genome Studies in Four Species of Calendula L. (Asteraceae)

Using Satellite DNAs as Chromosome Markers" by Samatadze, et al., who carried out the bioinformatic analysis of the C. officinalis DNA sequencing data, also clarified ploidy status and genome relationships of four Calendula species based on FISH mapping of 45S rDNA, 5S rDNA, and the identified satellite DNA families. This study provides a basis for the classification of Calendula L. (Asteraceae) . And the manuscript is well written and thoroughly described and it can find interest among the researchers in this field. However, there are some comments or questions before its possible publication as follows:

Answer: We thank the reviewer very much for the constructive comments and suggestions which are valuable in improving the quality of our manuscript. We have carefully revised the manuscript accordingly.

  1. The language used in this manuscript needs attentions, it should be edited by a native English-speaking expert.

Answer: The language has been improved throughout the manuscript (shown in blue). We hope that the English meets the Plants standards now.

  1. There are too many keywords, which need to be streamlined and avoided being the same as the title words. At the same time, I think it is necessary to use complete words when abbreviations appear for the first time in the text, like NGS, FISH.

Answer: This has been corrected. Lines 28-29.

  1. Figure 1 is not a good form of expression, it is unsightly and incomprehensible, so it is suggested to replace it with other drawing forms.

Answer: Figure 1 has been modified.

  1. In Table 1, all data need to be kept two decimal places.

Answer: This has been corrected.

  1. In Table 2, the format of the three-line table is incomplete, please modify and supplement it.

Answer: This has been corrected.

  1. The title of 2.2 is not specific enough to know the content of this part directly through the title.

Answer: The title of 2.2 has been modified.

  1. Although the manuscript cited more than 80 reference, more than half of them are outdated, which need to be updated and the preface and discussion part should be carefully revised. Answer: The references have been updated (shown in blue).